# The Human Virome and Its Crosslink with Glomerulonephritis and IgA Nephropathy

**DOI:** 10.3390/ijms24043897

**Published:** 2023-02-15

**Authors:** Fabio Sallustio, Angela Picerno, Francesca Montenegro, Maria Teresa Cimmarusti, Vincenzo Di Leo, Loreto Gesualdo

**Affiliations:** 1Department of Precision and Regenerative Medicine and Ionian Area, University of Bari Aldo Moro, 70124 Bari, Italy; 2Department of Interdisciplinary Medicine (DIM), University of Bari Aldo Moro, 70124 Bari, Italy

**Keywords:** virus, glomerulonephritis, virome, hepatitis, IgA nepropathy, HIV, immune system

## Abstract

The prokaryotic, viral, fungal, and parasitic microbiome exists in a highly intricate connection with the human host. In addition to eukaryotic viruses, due to the existence of various host bacteria, phages are widely spread throughout the human body. However, it is now evident that some viral community states, as opposed to others, are indicative of health and might be linked to undesirable outcomes for the human host. Members of the virome may collaborate with the human host to retain mutualistic functions in preserving human health. Evolutionary theories contend that a particular microbe’s ubiquitous existence may signify a successful partnership with the host. In this Review, we present a survey of the field’s work on the human virome and highlight the role of viruses in health and disease and the relationship of the virobiota with immune system control. Moreover, we will analyze virus involvement in glomerulonephritis and in IgA nephropathy, theorizing the molecular mechanisms that may be responsible for the crosslink with these renal diseases.

## 1. Introduction

Over the past ten years, research on the human gut microbiome has proliferated. This ecosystem is now considered for its role in metabolism, the immune system, and human health as a result of the enormous advancements in molecular technologies and newly created “omics” methodologies [1]. A number of metagenomic studies conducted in recent years have revealed a link between the microbial make-up of the human gut and a number of disorders, such as obesity, Crohn’s disease, and irritable bowel syndrome [2,3,4]. The gut microbiota contains at least 10^11^–10^12^ bacteria per gram of feces [5], and its composition varies with physiological factors such as geographic origin, age, dietary habits, malnutrition, and external factors such as probiotics or use of antimicrobial agents that can also upset the microbiota [6,7].

Human microbiota are made up of bacteria, yeasts, fungi, and also viruses, the makeup of which is still poorly understood. Research on pathogenic viruses indicates that bacteriophages, giant viruses, and plant-derived viruses are all present in large numbers in the human gut. The ability to reconstruct complete viral genomes from genetic material dispersed in the human gut using novel metagenomic techniques has opened up new avenues for research into the composition of the gut microbiome, its significance, and possible clinical applications [8].

The prokaryotic, viral, fungal, and parasitic microbiome exists in a highly intricate connection with the human host. This signals a paradigm shift. Recently, considerable attempts have been made to characterize the gut repertory; nonetheless, even for all microbes isolated from or found in the human gut, an effective repertoire survey is still required [5]. 

## 2. The Human Virome and Its Body Habitats

Members of the virome may collaborate with the human host to retain mutualistic functions in preserving human health. Evolutionary theories contend that a particular microbe’s ubiquitous existence may signify a successful partnership with the host. The higher risk of cervical cancer in women infected with high-risk Human Papillomavirus (HPV) strains is an example of how some persistent DNA viral infections are clearly related with disease. This may be comparable to the bacterial infections that persist in the microbiome and have the potential to cause disease but are not always manifest [9]. 

Although a person’s virome can alter over time, viral presence in a person can be extremely stable. Since viruses are present in the environment all the time, it is easy to imagine where the members of the changeable virome originate. However, recognized host defenses and underappreciated processes are probably the main components of the systems that support viral–host cohabitation. Viruses produce proteins that control the cell cycle [10], host gene expression], and host immunological responses [11,12,13,14]. Micro RNAs that control cellular functions are also encoded by viruses [15]. Consequently, viruses that are latent and persistently infected are constantly engaging with the host in a variety of ways. The intricate impacts of interactions between the virus and host during infection are being studied by many virologists. We are aware that viruses frequently target particular cellular pathways to control the cell and encourage viral replication, but each virus may employ a different strategy to accomplish the same control [15,16]. More research on the functional relationships between viruses and host cells will be carried out in the future on the human virome. 

The potential interactions between the viral and bacterial communities kept by complex organisms are an essential aspect of virome investigations. 

## 3. The Role of the Viruses in Health and Disease

### 3.1. Bacteriophages

Bacteriophages that attack bacteria are present in the human virome, and these phages might indirectly affect the host by changing the fitness and composition of infected bacteria. Due to the existence of various host bacteria, different anatomical regions may have rather varied phage compositions. Phages are widely spread throughout the human body. The effects of phage predation on the bacterial populations in humans are not well understood. Phage therapy, in which phages are purposefully administered to human patients to cure bacterial infections, provides a window on phage predation and host health. This strategy is becoming increasingly popular as drug-resistant bacteria start to appear [17]. Phage cocktails have recently been employed in numerous investigations to treat bacterial infection in a small number of patients, and their apparent efficacy [18] has inspired bigger clinical trials. Phages can transfer DNA between cells, giving bacterial genomes new functionality and potentially altering their fitness and pathogenicity [19]. Recent investigations on animals and in vitro have suggested that phages might directly interact with the host immune system. Without the involvement of bacteria, immunological responses can be triggered by phages via Toll-like receptor (TLR) signaling. Through the nucleotide-sensing receptor TLR9, *Lactobacillus, Escherichia, and Bacteroides* phages can promote the synthesis of Interleukin 12 (IL-12), IL-6, IL-10, and Interferon γ (IFN) [20,21]. The interactions between phages, bacteria, and the host immune system probably play significant roles in maintaining host immunological homeostasis (Figure 1). Phages can modify bacterial fitness and composition in a way that indirectly affects the host. Some phages and human cell viruses have the ability to integrate into the cells of their respective hosts, sometimes giving the host cells additional capabilities [22].

### 3.2. Eukariotic Viruses

Numerous investigations have found a high level of inter-individual variation in the human virome [23]. The virome, on the other hand, is typically quite consistent over time in a healthy adult, paralleling stability in the cellular microbiome. Virome instability is frequently linked to disease states. Virome populations have a variety of effects on their human hosts. Eukaryotic viruses that attack human cells generate immunological reactions, start infections, and occasionally even illness. 

When viruses are detected at locations with a local microbiota, they frequently include both viruses that replicate in the human cells there and viruses that infect the microbiota. It is only now beginning to be determined how much circulation exists between the sites. Often, respiratory viral infections can induce exacerbation of some glomerulonephritis such as IgA nephropathy (IgAN), causing onset of gross hematuria and hypertension. The reasons are not clear but may be partly due to mucosal immune system dysfunction [24,25,26,27]. It is unknown if small circular DNA viruses (*Anelloviridae* and *Redondoviridae*) that are common in respiratory samples and can also be discovered in feces emerge in the gut as a result of gastrointestinal tract replication or as a result of salivation [22]. *Anelloviridae* have been detected in greater frequency in immunocompromised patients, including as lung transplant recipients, Human Immunodeficiency Virus (HIV)-positive people, and people on immunosuppressive medications because of inflammatory bowel disease, indicating that they are generally under host immune control [28,29,30]. Another family of extensively occurring tiny circular DNA viruses that are typically identified in the respiratory system is the recently discovered *Redondoviridae*. They are viruses of the human oral cavity and respiratory tract associated with periodontitis and critical illness [31,32]. 

## 4. The Eukaryotic Virobiota and the Immune System Control

It is not a new concept that our body is colonized by bacteria residing in different body districts, including the skin, the gut [33,34], the reproductive and the respiratory system [35,36]. They are part of the human microbiota, whose composition has been widely explored during last decades, thanks to the development of modern sequencing tools and to the rapid growth of interest in different fields [37]. Concerning the use of new technologies, the advent of metagenomics has been crucial in obtaining new information about human microbiota and its composition. For instance, the metagenomics studies performed in the Human Microbiome Project (HMP) enabled the identification of a viral component in the human microbiota of healthy subjects, via the analyses of different samples collected from the nose, skin, mouth, vagina and stool [38]. 

This study identified several DNA viruses belonging to different families such as those of *Herpesviridae*, *Papillomaviridae*, *Polyomaviridae*, *Adenoviridae*, in addition to *Anelloviridae*, *Parvoviridae* and *Circoviridae*. Furthermore, more recent studies have been performed, identifying a huge community of viruses, both DNA and RNA viruses, co-habiting in different body surfaces, such as the skin, respiratory tract, oral/nasopharyngeal cavity and gastrointestinal tract, where their distribution is highly dependent on tissue district [22]. 

According to the canonical definition according to which viruses are typically defined as parasites, this finding may be quite confusing since they are commonly associated with disease and negative consequences.

Among viruses detected in human virobiota, the majority are formed by bacteriophages mainly placed in the gut, where 1015 particles were detected; they are temporally stable, with the capability of shaping gut bacterial taxa [39]. Furthermore, bacteriophages are more abundant than viruses that replicate and persist in eukaryotic cells [40]. Interestingly, even plant viruses have been identified, particularly in the feces, highlighting how diet can affect and alter the virobiota, the composition of which is dynamic [41].

Nonetheless, in this review we focused on eukaryotic viruses, especially in the relationship between viruses and the immune system, since it is well known that during the lifetime, humans host multiple animal viruses which often establish chronic infections. While bacteria residing in the gut and on the skin are known to be useful for the host, conferring metabolic and immune benefits, less is known about resident viruses and their relationship with eukaryotic cells [42,43]. 

Specifically, the stable presence of viruses infecting eukaryotic cells is an even more complex situation to study than those of bacteriophages [44]; in contrast to the classical serological methods that capture the infectious history of an individual at a given time, metagenomics may miss the detection of a virus that is no longer present in a patient or tissue. Additionally, chronic infections are often difficult to detect because certain viruses exist in a latent state and their level is too low for conventional methods. It is likely that the maintenance of a virus in the host could be useful for its interactions with both the microbiome and the host, contributing to the interindividual variation in immunity [45]. 

Hence, viruses probably behave somewhat like bacteria, which play a decisive role in shaping the immune system from the earliest years of life, together with the role of maintaining tissue homeostasis through an active microbiota-immune system crosstalk [46]. For instance, *Anelloviruses* (AV) are commensal viruses present in blood and other tissue, are not associated with any disease, and are thought to be part of the virus flora with a good influence on the host. Their abundance starts to increase after 12 months of age [47] and high levels of Torque Teno Viruses (TTVs), one of the main components of AV, were found in two-month-old babies [48], supporting the idea that they may influence the development of the immune system [46,49]. On the other hand, low levels of AV were detected in the blood of pregnant women whose children developed psychotic disorders, suggesting how a reduced AV content may be responsible for a strong activation of the maternal immune system, triggering psychosis in the progeny [50]. 

### The Eukaryotic Virobiota and the Immunodeficiency

Viruses can trigger or impair the immune response during life. For instance, Herpes Viruses (HV) are highly prevalent in the human population, especially *Herpes Simplex Virus 1* (HSV1), *Human Cytomegalovirus* (HCMV), *Epstein-Barr Virus* (EBV) and *Varicella Zoster Virus* (VZV). HSV, a virus normally acquired in childhood, triggers a delicate interplay between innate and adaptative immune response: it activates natural killer cells and dendritic cells to produce Type I Interferon (IFN), mainly IFNα and IFNβ, mediating a protective role against disease, both in mouse models and in human studies [51]. After the first infection, HSV is maintained in a latent state that can be reactivated over one year; nonetheless, the chronic infection has been demonstrated to have a protective role against viral and bacterial infections, such as those against γ-herpes viruses [52]. On the other hand, *Simian Immunodeficiency Virus* (SIV) is associated with strong expansion of intestinal virobiota in rhesus monkeys, driving immunodeficiency and causing AIDS [40]. 

The neutral, beneficial, or deleterious effects on the host are often the result of a fine balance between a viral component and the host immune system. This is the case of EBV, responsible for the “kisses disease” with a high incidence in non-immunocompromised young people. It is carried on as an asymptomatic latent infection of the B lymphoid system. Since it results from T cell response control, people with a reduced T cell count (immunodeficiency or immunosuppression) have an increased risk of B lymphoproliferative diseases such as Burkitt Lymphoma, Hodgkin Lymphoma or diffuse large-B-cell lymphoma [53]. 

In conclusion, it is not possible to generalize about the effects that human virobiota exert on the immune system; this is because of the partial knowledge about the assortment of all viral species that colonize our organism, but also because of the fluidity of the relationships between viruses and the immune system, where the balance of the virus-host relationship is essential in determining health consequences (Figure 1). The picture is then complicated by the variability of individual virus behavior, considering that some potentially harmless viruses can be reactivated after years, as well as by the existence of a population of viruses, whose effects on human health are not yet fully known.

## 5. Virus Infection and Nephropathy

Viral infections are involved in several glomerular diseases, but the pathogenetic links between viral infection and kidney disease are often difficult to establish. The known mechanisms differ for each different viral nephropathy [54]. Generally, acute glomerulonephritis, a viral infection of the glomerulus, causes the local release of cytokines which leads to glomerular cell proliferation [55]. This acute nephropathy resolves spontaneously if the viral infection is rapidly cleared. Chronic glomerulonephritis results from the continuous formation of immune complexes, due to persistent viral infection. Additionally, viral proteins can exacerbate inflammatory conditions and worsen glomerulopathy. Evidence suggests that several viruses such as HIV-1 and hepatitis viruses can cause glomerular damage after both acute and chronic infection. As for cases associated with other viral infections, such as *parvovirus* B19 and *cytomegalovirus* (CMV), the mechanisms remain incompletely understood.

## 6. Hepatitis B and C and Glomerulonephritis

Glomerulonephritis (GN) is a crucial cause of renal failure. It includes several immune-mediated disorders that trigger inflammation in the various compartments of the kidney, inducing intraglomerular inflammation and cell proliferation associated with hematuria [56,57,58]. 

Various pathogenic processes are implicated in the induction and progression of glomerular inflammation: antibody deposition, the cell-mediated immune mechanism, complementary activation, hemodynamic alterations, influx of circulating leukocytes, cytokines, and growth factor synthesis [56].

Interestingly, several studies have also shown the crucial role of infections in causing glomerulonephritis [59] and confirmed the main role of extrahepatic infection of the hepatitis B (HBV) and C (HCV) viruses [60,61] (Table 1). Furthermore, other works highlighted the relationship between occult HCV/HBV infections and GN. In fact, occult HCV has been detected in up to 50% of patients with idiopathic membranous nephropathy, IgA, Focal Segmental Glomerulo Sclerosis (FSGS), Antineutrophilic cytoplasmic antibody (ANCA) positive vasculitis and membranoproliferative GN [62], while occult HBV has been identified in selected cases of idiopathic membranous nephropathy and IgA, where viral antigens were found in renal tissue, in absence of viremia [63]. This evidence emphasized an association with GNs, even when the virus is not replicating.

Hepatitis B represents the most common chronic viral infection in the world [63,64]. In the countries where HBV infection is more common, there is also a higher percentage of HBV-induced glomerulonephritis (HBV-GN), further confirming the link between the virus and the disease [64,65,66].

HBV- or HCV-associated glomerulonephritis can be diagnosed by evaluating the presence of positive viral markers in serum and, through immunohistochemistry, by verifying the presence of viral antigens in a renal biopsy specimen [67,68,69]. In fact, glomerulonephritis is induced by immune complexes such as HBV surface antigen (HBsAg) and core antigen (HBcAg), which tend to accumulate in the glomerular mesangium and in the endothelial cells [64,65,66] (Table 1).

According to the HBV or HCV infection, different kinds of infection-associated glomerulonephritis have been identified, such as membranous nephropathy (MN), which is the most common pathological diagnosis, followed by mesangial proliferative glomerulonephritis (MNPG), IgA nephropathy (IgAN) and immune complex-related vasculitis (polyarteritis nodosa glomerulonephritis, PAN) [70,71]. The main features, both immunological and clinical, are summarized in the following Table 1.

**Table 1 ijms-24-03897-t001:** Main features of HBV and HCV- associated Glomerulonephritis.

Virus-Associated Glomerulonephritis	Virus Type	Immunological Traits	Clinical Features
MN	HBV	HBsAg, HBcAg, or HBeAg immune complex with IgG deposition at subepithelial level	Lower levels of circulating complement; stronger segmental glomerular damage, mesangial cell proliferation and tubulointerstitial damage [72].
MPGN	HBV HCV	HBsAg- IgG immune complex	Reduced serum C3 levels; nephritic/nephrotic syndrome. Lobular appearance of the glomerulus with cleavage of the basement membrane and mesangial, subendothelial and subepithelial deposits [73,74].
MC	HBV	HBs-IgG or IgM immune complex	Severe renal manifestations involving nephrotic-range proteinuria and acute kidney disease (AKI) [64].
PAN	HBV HCV	HBsAg-related immune complex	Vascular immune complex deposition

HBV-induced MN presents histological deposits of subepithelial immune complexes, visible with the electron microscope. Patients with HBV-induced MN have been observed to have lower levels of circulating complement and more frequent segmental glomerular damage, mesangial cell proliferation, and tubulointerstitial damage on histology, compared to subjects affected by idiopathic MN [73] (Table 1).

In addition to MN, MPGN is the second-most common renal glomerular syndrome in HBV carriers and is induced by HBsAg-related immune complex. Clinically, it presents with nephritic syndrome or nephrotic syndrome and reduced serum C3 levels [71,72].

Patients may also have a protracted course of HBV-related mixed cryoglobulinemia (MC), often with severe renal manifestations involving nephrotic-range proteinuria and acute kidney disease (AKI) [59].

PAN glomerulonephritis in HBV patients is the result of HBsAg antibody immune complexes depositing in medium-sized blood vessels. HBV-associated PAN leads to AKI as well as to heart attack. However, thanks to vaccination against HBV, both cases of NM associated with HBV in children and cases of HBV PAN have been reduced [59]. These observations allow us to reflect on the usefulness of vaccinations also to prevent less frequent virus-related diseases. Less is known about HCV-related GN, but the most correlated glomerular disease is the immune complex-mediated MPGN [69,74,75].

Depending on the type of disease, a different therapy may be administered. Treatment of HBV-related glomerulopathies is essentially antiviral. In particular, the treatment of HBV-related MPGN with type 3 cryoglobulinemia is targeted toward the control of HBV viremia with the use of direct-acting antivirals, rituximab, plasmapheresis, and possible cytotoxic therapy, as systemic cryoglobulinemic vasculitis are unusual with HBV. Steroids, with the exception of HBV-associated polyarteritis nodosa, have proved to be ineffective, while immunosuppressants are associated with a high risk of HBV infection exacerbation [70]. The treatment of HCV-related glomerulonephritis includes conventional or innovative immunomodulatory drugs, possibly in combination with antivirals [75].

These infection-related glomerulonephritis are quite severe, and viral antigens should be detected in patients’ kidney tissue to determine the correct diagnosis for the therapy to be used.

## 7. Human Immunodeficiency Virus (HIV) and Glomerulonephritis

Common consequences of Human Immunodeficiency Virus (HIV) infection are chronic diseases in several organs, such as the kidneys, in which the virus determines direct and indirect damage. The first one is related to the cytopathic effects of the virus within the renal parenchymal cells, disrupting normal cell activity. Indirect injuries are connected to the response of the immune system to HIV infection, including the formation of immune complexes depositing in the kidneys [76].

In February 2018, the Kidney Disease Improving Global Outcomes (KDIGO) has classified HIV nephropathies according to the tissue compartment involved in glomerular tufts, tubules, capillary vessels, and others. The main groups of glomerulonephritis associated with HIV infections are HIV-associated nephropathy (HIVAN) and intraglomerular immune complex deposition (HIVICK) [75].

HIVAN is a well-defined clinicopathologic entity manifesting as collapsing glomerulopathy associated with a consistent tubulointerstitial disease or, in its attenuated form, to focal segmental glomerulosclerosis (FSGS) [77].

HIVICK includes a spectrum of renal diseases, among them, membranous nephropathy, membranoproliferative glomerulonephritis, lupus-like nephritis, and IgA nephropathy (IgAN) [78].

It is difficult to establish a clear causal link between these glomerulonephritis and viral infection; thus, the diagnosis of immune complex kidney disease in the setting of HIV remains plausible only when other secondary causes have been excluded.

Since renal cells do not express conventional HIV receptors (CD4, CXCR4, and CCR5) [79], some authors have suggested that the transfer of viral material from HIV-infected T-lymphocyte to tubular epithelial cells require stable cell–cell contact and occur independently of CD4 expression. Other evidence has shown that nonconventional receptors could be used. For example, C-type lectins seem capable of binding HIV independently of CD4; DEC-205 receptor is able to recognize the carbohydrate structures present on cellular and viral proteins and has been suggested to play the role of HIV receptor in tubular cells [80]. Furthermore, for the entry of HIV into podocyte cells, the possible mechanism suggested the use of lipid rafts [81].

The interaction between HIV infection of kidney epithelial cells and the apolipoprotein 1 (APOL1) has been recognized to be essential in HIVAN pathogenesis [82] since 2010, especially among the African population carrying variants of the APOL1 allele, where this disease is observed in 90% of cases [83].

In fact, besides the wild type allele G0, APOL1 gene has two variants named G1 and G2 that are indicated as Renal Risk Variants (RRVs) [82]; about 13% of African Americans carry two high-risk alleles (either G1/G1, G2/G2, or G1/G2), putting them at a 3- to 30-fold increased risk of developing future kidney disease. While G0 heterozygotes demonstrate relative protection from developing kidney disease, most people with APOL1 RRV alleles do not develop kidney disease [84]. The current understanding is that an environmental trigger is required to cause disease development. The innate immune response to HIV upregulates APOL1 by type 1 interferons (IFN) signaling via different mechanisms such as the stimulator of interferon genes (STING) mediation, the upregulation of retinoic acid-inducible gene I (RIG-I) or those of Retinoic Acid-Inducible Gene I (RIG-I), that mediates the transcriptional activation of IFN [85].

Since no other viruses causing similar kidney disease in patients with APOL1 RRVs has been identified, further factors, highly related to HIV infection, should participate in APOL1 upregulation. In vitro studies with HIV-expressing and HIV-infected podocytes showed that the expression of viral protein Nef increases the characteristic podocyte proliferation and dedifferentiation in HIVAN glomerular disease [86]. Nef forms direct protein–protein interactions with Src family kinases activating the pathway of STAT3 which binds the APOL1 promoter [87,88].

Further studies revealed that APOL1 RRVs induced an autophagy blockade in podocytes by multiple mechanisms [83,89,90], as listed in Table 2.

In contrast to HIVAN, less is known about HIVICK. It is a not well defined, long-term progression renal disease associated with HIV infection, with aberrant immune regulation and increased gamma globulin contributing to immune complex deposition.

The formation and deposition of the immunocomplex in capillary loop and mesangium determine proteinuria, hematuria and decreased glomerular filtration rate. These conditions commonly occur when hypergammaglobulinemia is associated with the polyclonal proliferation of B cells due to viral coinfection [78].

Viral antigens deriving from HIV replication in the kidney, including p24 (capsid) and gp120 (envelope) antigens, can trigger the production of broadly neutralizing antibodies that can determine immunocomplex formation, developing the disease.

## 8. Parvovirus B19, Cytomegalovirus, and Glomerulonephritis

Human parvovirus B19 is a ubiquitous nonenveloped single-stranded DNA virus with exclusive human hosting. It has been associated with various cases of kidney injuries with different glomerular phenotypes. For example, it was first associated with glomerulonephritis in seven patients with sickle cell disease, in whom acute infection and the consequent aplastic crises were associated with segmental proliferative glomerulonephritis and, at a later stage, with FSGS [93,94].

In immunocompromised individuals, insufficient production of neutralizing antibodies can lead to chronic manifestations of parvovirus B19. Indeed, despite the presence of neutralizing antibodies, parvovirus B19 DNA can be detected for years in the bone marrow and peripheral blood [95].

*Cytomegalovirus* (CMV), a *double-stranded DNA virus* [96], is one of the major viral pathogens responsible for tubulointerstitial nephritis in renal allograft recipients [97], and a probable inducer of collapsing FSGS. CMV-associated glomerular diseases have shown a direct role in viral infection by including cytomegalic inclusion bodies and/or viral particles within glomerular cells [94]. CMV is known to use microRNAs (miRNAs) and accessory proteins in mimicry strategies, like many other viruses with large DNA genomes and a nuclear replication cycle [98].

The involvement of miRNAs in infection is due to their lack of antigenicity and their ability to post-transcriptionally inhibit the expression of specific mRNA species. Thus, miRNA may allow the virus to modify host functions [99]. In fact, CMV encodes several host mimetic miRNAs that are expressed during its lytic phase to suppress host proteins involved in antigen presentation [100]. CMV produces the viral accessory protein UL38, which can manipulate the signaling of the cellular mammalian target of rapamycin mTORC1, to facilitate viral protein production and suppress autophagy and apoptosis to promote viral persistence [99]. However, beyond these hypotheses, the pathological molecular mechanisms for the CMV and B19 virus, in glomerulonephritis, remain unclear.

## 9. IgA Nephropathy and Viruses

Immunoglobulin A nephropathy (IgAN), also known as Berger’s disease, is one of the common forms of glomerulonephritis and remains the main source of mortality and morbidity in glomerulonephritis. It is caused by the deposition of IgA immunoglobulin in the glomerular basement membrane. Usually, it manifests in the second and third decades of life, with a higher incidence in males, and it is more common in whites than in blacks with an advanced incidence in Asians compared to Caucasians [101].

IgAN can lead to end-stage renal disease, and this occurs in 20–40% of IgAN patients within 20 years of biopsy diagnosis [102]. The diagnosis can be proved through pathological evaluation via invasive kidney biopsy [102]. Nevertheless, the correct etiology is not entirely complete and needs to be investigated [103].

IgA disease develops through an interaction between genetic, epigenetic and environmental factors, which lead to the formation of glomerular deposits of IgA, especially polymeric IgA1 [104,105,106,107]. Since the presence of poorly galactosylated IgA1 O-glycoforms is not sufficient to cause IgA nephropathy, the “multi-hit” hypothesis has recently prevailed. This includes four hits [108,109]: one-hit is the presence of high levels of IgA1 deficient in galactose, the Gd-IgA1, responsible of cellular proliferation and high production of the extracellular matrix, cytokines, and chemokines, that lead to glomerular damage [102]. These Gd-IgA1 are identified and struck by autoantibodies (two-hit); these antibodies can occur as a result of recurrent mucosal infections; in IgAN patients there are elevated levels of glycan-specific IgG, related to proteinuria, which possess a variable region of the heavy chain that recognizes galactose-deficient IgA1. This event leads to the formation of circulating immune complexes (CIC) (three-hit), which can settle in the glomerular mesangial area, causing inflammatory reaction and tissue damage (four-hit) [110,111]. Kidney injury following the immune complex deposition is characterized by local inflammation, complement activation, cell proliferation, and fibrosis [108].

Furthermore, in humans, the pathogenesis of IgAN is also associated with intestinal homeostasis, as the lymphoid tissue associated with the intestine is the primary source of IgA. Recently, it has been shown that IgAN patients had greater levels of the BAFF cytokine in their serum, which was correlated with higher concentrations of five particular microbiota metabolites and higher levels of the APRIL cytokine. Additionally, compared to HS, IgAN patients have more circulating gut-homing (CCR9+ 7 integrin+) regulatory B cells, memory B cells, and IgA+ memory B cells; both total plasmablasts (PBs) and intestinal-homing PBs were present in high amounts in IgAN patients [112]. These findings support the idea that intestinal mucosal hyperresponsiveness plays a pathogenic role in IgAN by inducing a substantial difference in the number of intestinally activated B cells between IgAN patients and HSs. Therefore, the complex pathophysiology of IgAN may be influenced by a number of mechanisms, including the intestinal-renal axis.

Lately, the concept of a human virome has aroused a lot of interest. Viromes represent one of the most variable components of the human intestinal microbiome, which changes from childhood to adulthood, and consist of both the viral component of the microbiome, dominated by bacteriophages, and a variety of DNA viruses that directly infect eukaryotic cells [38,39,113]. The virome is influenced by responses to different environments, lifestyles or infections. Viruses enter into the human body using mucous surfaces, where they interact with the host’s immune defense, including commensal bacteria [103].

There are several hypotheses regarding the role of infections in the pathogenesis of IgA nephropathy, including the main role of pathogens, chronic exposure to mucosal infections, abnormal alterations of commensal microbiomes [114].

Infections activate the immune system of the mucosa causing it to produce IgA, which is therefore produced in excess, causing deposits. In patients with recurrent tonsillitis, granular depositions of *Adeno* and *Herpes Simplex* viral antigens have been identified in the mesangial glomerular areas [115]. Moreover, after tonsillectomy, the serum levels of Gd-IgA1 reduced, thus indicating the tonsils as a source of poorly galactosylated polymeric IgA1 [116].

Indeed, clinically, a high percentage of IgAN patients are known to suffer from respiratory infections, the most common of which are tonsillitis and pharyngitis. Consequently, patients may be weaker and therefore vulnerable to hematuria, proteinuria or deterioration of renal function [106].

IgAN-related Cytomegalovirus or Toxoplasma infections have been also observed [115]. Moreover, antigens of *Haemophilus parainfluenzae* and *Staphylococcus aureus* have been found in renal tissue of IgAN patients more frequently compared to other renal diseases [117]. Furthermore, antigens of Human Cytomegalovirus, Adeno and Herpes simplex virus, Hemophilus parainfluenza, Staphylococcus and Epstein-Barr virus have been detected in renal tissues from patients with IgAN, together with IgA deposits [102,117,118].

Current studies have also reported the connection between HBV infection and IgA nephropathy; clinicopathological characteristics and outcomes of IgAN patients with HBV infection are different compared to patients without infection, suggesting that HBV is a risk factor for the progression of IgAN [119].

Different studies have tried to explain the connection between Hepatitis C and IgAN, but the investigation in this specific field needs to be improved [120]. A case has been reported of a 16-year-old patient with oliguria and development of anasarca for 5 months, followed by proteinuria. Diagnostic tests revealed an HCV infection, and the kidney biopsy also showed the presence of IgA deposits [120]. Other studies reported patients with IgAN and HCV, that after 24 weeks of antiviral treatment showed reduction of proteinuria [121]. Therefore, these cases suggest that hepatitis C may also cause IgAN.

From all these studies, the need to deeply study the molecular mechanisms by which the virome may affect IgAN emerges.

Recent studies may provide new insights into biological mechanisms linking viral infections to IgAN.

A whole-genome DNA methylation screening in CD4+ T cells from IgAN patients identified a hypermethylated region comprising Vault RNA 2-1 (VTRNA2-1), a non-coding RNA also known as the precursor of miR-886 (Pre-mi-RNA) [122].

VTRNA2-1 is a direct inhibitor of the protein kinase R (PKR), whose phosphorylation activates CREB, a classical cAMP-inducible CRE-binding factor interacting with a region of the IL-6 promoter, leading to IL-6 production. Recently, the involvement of interleukin 6 in IgAN as crucial for glomerular immunoglobulin A deposition and the advancement of renal pathology has been established [123].

Our recent data show that the VTRNA2-1/PKR/CREB/IL-6 pathway is upregulated in IgAN patients and that it is responsible for the elevated IL-6 levels characterizing the disease [124].

Bacterial and viral RNA may therefore further trigger the PKR/CREB/IL-6 pathway, already activated in IgAN patients due to the epigenetic silencing of the VTRNA2-1 PKR inhibitor, leading to an overload of IL-6 production. It was shown that human B cells infected in vitro with Epstein-Barr virus (EBV) secrete galactose-deficient IgA1 and that IgAN patients had more lymphoblasts/plasmablasts that were surface-positive for IgA, infected with EBV, and displayed increased expression of homing receptors for targeting the upper respiratory tract [125]. Another study showed instead that PKR is regulated by EBV RNA [124]. These studies support therefore the involvement of viruses in IgAN pathogenesis and in the regulation of the PKR/CREB/IL-6 IgAN pathway.

COVID patients who showed kidney complications were also examined [126]. Biopsies frequently showed acute tubular injury, as well as glomerular nephropathy such as collapsing glomerulopathy.

Several cases of IgAN patients that probably developed the disease following COVID-19 or following the vaccination have been reported. The causes, according to this study, may be due to the role of the IgA-mediated immune response, evidenced by early seroconversion to IgA in patients with COVID-19 and the role of IgA in immune hyperactivation [127]. Nevertheless, this link needs to be further explored.

The connection between COVID-19 vaccination and IgAN has also been explored; in some cases, after the COVID-19 vaccine, patients already affected by IgAN showed recurrence of significant hematuria and proteinuria. Some patients had gross hematuria after the first dose of the vaccine, accompanied by increased proteinuria, arthralgia, and abdominal pain [128,129]. These data indicate that the connection between anti-COVID vaccination and IgAN should be investigated and that it is necessary to monitor those who show symptoms of IgA disease after the vaccine or to check those who are already affected by the disease and need to undergo vaccination.

However, these data may further support the involvement of the PKR in IgAN through its binding to COVID 19 RNA [124].

Intensifying the studies on correlation between IgAN and viruses is needed to better understand the disease, to increase the therapeutic options and to improve patient treatment.

## 10. Conclusions

Each person harbors unique and diverse viral communities, according to research on the diversity and makeup of the human virome population at various bodily sites (Figure 1). Future research is required to better understand the DNA and RNA viromes at various anatomical sites and to connect changes in viral makeup to glomerulonephritis. It is yet unclear how early viral colonization affects long-term health effects, which calls for cautious research.

Some viruses appear to be long-term “passengers” or “commensals,” but they actually participate in benign colonization and cannot be linked to any specific disease.

Even if this review does not cover all the possible connections between the virome and health and disease states, emerging information suggests that many factors that affect the human microbiome also frequently affect the virome, making it difficult to separate the effects of each. In addition to actively engaging with other microbes, resident viruses also do so with the immune system of mammals. Numerous exciting findings have thus far only been attained in studies using animals or in vitro investigations, drawing attention to the need for studies involving humans. From analyzed data, many crosslinks between the virome and glomerulonephritis emerged. Moreover, new connections between viruses and IgAN, the most common form of glomerulonephritis worldwide, are being discovered, explaining some aspects of disease not understood so far. The huge world of the human virome is starting to be comprehended, setting the foundation for a variety of important future studies.

## Figures and Tables

**Figure 1 ijms-24-03897-f001:**
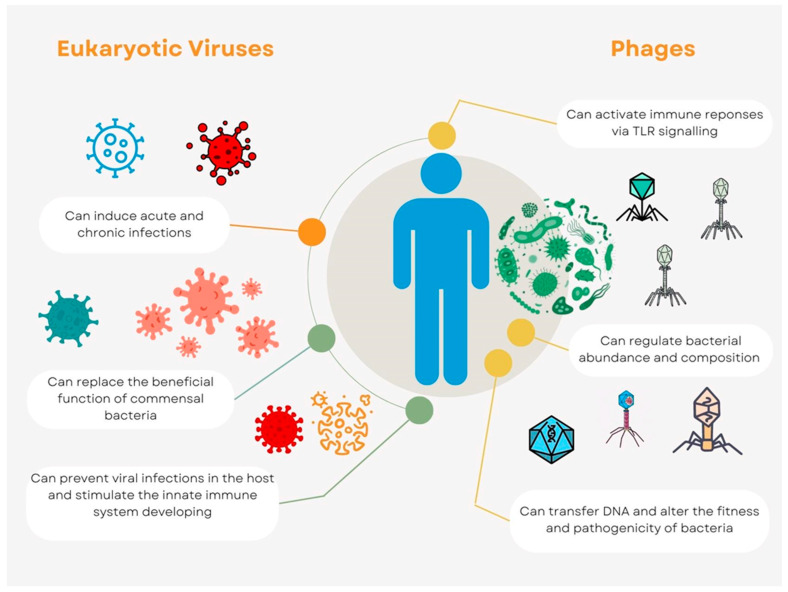
Virome interconnections with host. The host health is impacted by eukaryotic viruses in both negative and positive ways (green and orange lines, respectively). Phages interact with the host through the bacterial population that is associated with it, and these interactions, either directly or indirectly, could have unknown implications (yellow lines) on human health.

**Table 2 ijms-24-03897-t002:** Main driving autophagy mechanisms mediated by APOL1 RRVs.

APOL1 RRVs-Dependent Autophagy Blocking Mechanisms in Podocytes
Inhibition of autophagosomes maturation via stimulating the formation of Rubicon- UV Radiation Resistance Associated gene protein (UVRAG) complexes.
Inhibition of lysosome reformation due to the accumulation of autophagolysosomes, reducing mTOR expression [91].
Upregulation of miR-193a expression, a known inducer of oxidative stress that stimulates cell death in podocytes, destabilizing adherens junction, disorganizing the actin cytoskeleton and blocking autophagy [91]. As a result, it causes decreased assembly of the PI3KC3-autophagy-related protein 14L and PI3KC3-UVRAG complexes needed for nucleation and maturation of autophagosomes [92].

## Data Availability

Not applicable.

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
