# Peer review of "The Human Virome and Its Crosslink with Glomerulonephritis and IgA Nephropathy"

_ijms, 2023, doi:10.3390/ijms24043897_

Round 1
Reviewer 1 Report
Dear Authors,
please find below my review of your manuscript. The role of the human viriome is very important issue, that needs better understanding in many fields of medicine.
The manuscript looses focus on the main research aim in many places, and would need to be rewritten.
Abstract: the introduction in the abstract section is too long, and contains the information’s that are not focussing on what actually the researchers will discuss in their work. Please rewrite.
Introduction: lines 43-49 with only one citation? Many statements not confirmed by citations
Chapter 2:
line 59: “perhaps”?
line 61 “In order to find viruses”? What does it mean?
Lines 61/62: the techniques are provided without the line of thought. Please subgroup the techniques specifically (which PCR? why not NGS? how about the cell culture?)
In total, this chapter does not represent what is shown in the title of the manuscript. I would cancel from lines 55 till line 90 the sentences, because they do not present the concrete information about the topic, and authors can easily present their work without this part.
Chapter 3:
The information provided are not turning right. If you declare to study in this part The role of the viruses in health and disease, why you start by speaking about the bacteriophages? If you want to do so, please do not mix and match, but provide the division: viruses, and the phages, since you also have provided this division in your image in this review.
Cancel the sentence in line 114.
If you’re focusing on the viriome from the glomerulonephritis, why you discuss the examples about the upper respiratory tract? This part is mostly reflected in all the work provided in the manuscript. It generally need to be re-written because all the chapters loose focus. Please focus on what you wanted to present in the chapters, and stick to the topic.
One god idea would be to provide the separate subsection in each chapter about the immunodeficiency whenever it is stated by the authors, since it represents surely one very interesting and important research question about the changed viral environment.
from the chapter 5: The manuscript is too long, and not focussed. I would suggest that some of the content from the further chapters be provided in the tables not as text, to avoid such long pages. That will shorten the pages.
Conclusion should be succinct and shorter. Based on the conclusion one can declare what was the purpose of the manuscript, and how the researchers have found their way in it.
Details: avoid double spaces and please provide spaces before the brackets informing about the specific references. Please provide the Image in previous pages, since in the end it is not receiving enough attention. I would suggest after the Introduction. Please do not use the common language in the manuscript.
Best,
Reviewer.
Author Response
We thank the reviewer for his very helpful and valuable comments and suggestions. We have rewritten large part of the manuscript following the reviewer indications. We think that thanks to his suggestion the manuscript has been now improved.
We have attached a “Tracked changes manuscript” file with changes from the original paper highlighted in the red font to follow changes.
Point 1: Abstract: the introduction in the abstract section is too long, and contains the information’s that are not focussing on what actually the researchers will discuss in their work. Please rewrite.
Response 1: We have now rewritten the abstract, focussing on what we discuss in the paper.
Point 2: Introduction: lines 43-49 with only one citation? Many statements not confirmed by citations
Response 2: We have now referenced the different statements
Point 3: Chapter 2:
line 59: “perhaps”?
it was wrong, we mean “like”, we have now corrected the sentence
line 61 “In order to find viruses”? What does it mean? Lines 61/62: the techniques are provided without the line of thought. Please subgroup the techniques specifically (which PCR? why not NGS? how about the cell culture?).
In total, this chapter does not represent what is shown in the title of the manuscript. I would cancel from lines 55 till line 90 the sentences, because they do not present the concrete information about the topic, and authors can easily present their work without this part.
Response 3: As suggested we have cancelled the sentences from lines 55 till line 90.
Point 4: Chapter 3:
The information provided are not turning right. If you declare to study in this part The role of the viruses in health and disease, why you start by speaking about the bacteriophages? If you want to do so, please do not mix and match, but provide the division: viruses, and the phages, since you also have provided this division in your image in this review.
Response 4: The human virome comprises bacteriophages that infect bacteria and can affect the host indirectly via modulating of bacterial composition and bacterial fitness (as reported for example in “Nature reviews, The human virome: assembly, composition and host interactions”). Therefore, we would briefly discuss this topic, even because the relative chapter is entitled “The role of the viruses in health and disease”. We have now better explained this concept and, as suggested by the reviewer, we have provided a better division between phages and the viruses.
Point 5: Cancel the sentence in line 114.
Response 5: We have cancelled the sentence in line 114
Point 6: If you’re focusing on the viriome from the glomerulonephritis, why you discuss the examples about the upper respiratory tract? This part is mostly reflected in all the work provided in the manuscript. It generally need to be re-written because all the chapters loose focus. Please focus on what you wanted to present in the chapters, and stick to the topic.
Response 6: Often, respiratory viral infections can induce exhacerbations of some glomerulonephrithys such as IgA nephropathy (IgAN), causing onset of gross hematuria and hypertension. That’s why we discuss the examples about the upper respiratory tract. We have now explained this issue (lines 164 -167 of the tracking change document) and we have re-written some parts to remain focused.
Point 7: One god idea would be to provide the separate subsection in each chapter about the immunodeficiency whenever it is stated by the authors, since it represents surely one very interesting and important research question about the changed viral environment.
Response 7: We thank the reviewer for this suggestion. When applicable, we have provided in the chapters a subsection about the immunodeficiency.
Point 8: from the chapter 5: The manuscript is too long, and not focussed. I would suggest that some of the content from the further chapters be provided in the tables not as text, to avoid such long pages. That will shorten the pages.
Response 8: From the chapter 5 to the chapter 9 we began to discuss deepenly the principal topic of the Review (human virome and its crosslink with glomerulonephritis) but we realized and agree with the reviewer that the manuscript is too long. As suggested, we have now shortened the content and we have summarize part of text in two Tables.
Point 9: Conclusion should be succinct and shorter. Based on the conclusion one can declare what was the purpose of the manuscript, and how the researchers have found their way in it.
Response 9: We have shortened and focused the conclusions on the manuscript purpose.
Point 10: Details: avoid double spaces and please provide spaces before the brackets informing about the specific references. Please provide the Image in previous pages, since in the end it is not receiving enough attention. I would suggest after the Introduction. Please do not use the common language in the manuscript.
Response 10: We have revised spaces, language and, as suggested, we have moved the Figure after the introduction.

Reviewer 2 Report
The human virome and its crosslink with glomerulonephritis 2 and IgA nephropathy
General comments
Sallustio and colleagues present a survey of work on the human virome and highlight the role of the viruses in health and diseases, particularly focusing on glomerulonephritis and IgA nephropathy. The review is well written, although there is room for improvement. The authors can address the specific concerns below to improve the review further.
Specific comments
Sentences in lines 138 to 140 require citations
Sentence in line 141 beginning “Some phages may also directly interact with …” is incomplete
Sentences in lines 142-149 require citations
What were the limitations of this review?
The authors can consider restricting the style of the paragraphs – some paragraphs have just one sentence
The comments are attached, herewith

Author Response
We thank the reviewer for his very helpful and valuable comments and suggestions. We think that thanks to his suggestion the manuscript has been now improved.
We have attached in Supplemental Materials a “Tracked changes manuscript” file with changes from the original paper highlighted in the red font to follow changes.
Point 1: Sentences in lines 138 to 140 require citations
Response 1: We have now included the citation.
Point 2: Sentence in line 141 beginning “Some phages may also directly interact with …” is incomplete
Response 2: We have fixed this error.
Point 3: Sentences in lines 142-149 require citations
Response 3: As suggested by Reviewer 1 the sentences have been splitted in a subsection of the paragraph and we have now included the citations.
Point 4: What were the limitations of this review?
Response 4: We have added in the conclusion the limitation that this review does not cover all the possible connections between the virome and health and disease states (line 680).
Point 5: The authors can consider restricting the style of the paragraphs – some paragraphs have just one sentence
Response 5: Several paragraphs have been restricted as suggested by Reviewer 1 and we have also restricted the style of the paragraphs.
The comments are attached, herewith
